# *In silico* investigation of the mechanisms underlying atrial fibrillation due to impaired Pitx2

Jieyun Bai[1,2]*, Andy Lo[2], Patrick A. Gladding[3], Martin K. Stiles[4], Vadim V. Fedorov[5], Jichao Zhao[2]*

**1** Department of Electronic Engineering, College of Information Science and Technology, Jinan University, Guangzhou, China, **2** Auckland Bioengineering Institute, University of Auckland, Auckland, New Zealand, **3** Department of Cardiology, Waitemata District Health Board, Auckland, New Zealand, **4** Waikato Clinical School, Faculty of Medical and Health Sciences, University of Auckland, New Zealand, **5** Department of Physiology & Cell Biology and Bob and Corrine Frick Center for Heart Failure and Arrhythmia, The Ohio State University Wexner Medical Center, Columbus, Ohio, United States of America

* bai_jieyun@126.com (JB); j.zhao@auckland.ac.nz (JZ)

**Data Availability Statement:** All computer codes and cellular models are available from the repository CellML (http://models.cellml.org/

## Abstract

Atrial fibrillation (AF) is the most common sustained cardiac arrhythmia and is a major cause of stroke and morbidity. Recent genome-wide association studies have shown that paired-like homeodomain transcription factor 2 (Pitx2) to be strongly associated with AF. However, the mechanisms underlying Pitx2 modulated arrhythmogenesis and variable effectiveness of antiarrhythmic drugs (AADs) in patients in the presence or absence of impaired Pitx2 expression remain unclear. We have developed multi-scale computer models, ranging from a single cell to tissue level, to mimic control and Pitx2-knockout atria by incorporating recent experimental data on Pitx2-induced electrical and structural remodeling in humans, as well as the effects of AADs. The key findings of this study are twofold. We have demonstrated that shortened action potential duration, slow conduction and triggered activity occur due to electrical and structural remodelling under Pitx2 deficiency conditions. Notably, the elevated function of calcium transport ATPase increases sarcoplasmic reticulum $Ca^{2+}$ concentration, thereby enhancing susceptibility to triggered activity. Furthermore, heterogeneity is further elevated due to Pitx2 deficiency: 1) Electrical heterogeneity between left and right atria increases; and 2) Increased fibrosis and decreased cell-cell coupling due to structural remodelling slow electrical propagation and provide obstacles to attract re-entry, facilitating the initiation of re-entrant circuits. Secondly, our study suggests that flecainide has antiarrhythmic effects on AF due to impaired Pitx2 by preventing spontaneous calcium release and increasing wavelength. Furthermore, our study suggests that $Na^+$ channel effects alone are insufficient to explain the efficacy of flecainide. Our study may provide the mechanisms underlying Pitx2-induced AF and possible explanation behind the AAD effects of flecainide in patients with Pitx2 deficiency.

workspace/5c7) and GitHub (https://github.com/aspirerabbit).

**Funding:** This work was supported by the National Natural Science Foundation of China (61901192), the National Key R&D Program of China (No.2019YFC0120100 and 2019YFC0121900), Health Research Council of New Zealand (JZ), and NIH HL135109 and HL115580 (VVF). The funders had no role in study design, data collection and analysis, decision to publish, or preparation of the manuscript.

**Competing interests:** The authors have declared that no competing interests exist.

## Author summary

Atrial fibrillation (AF) is an abnormal heart rhythm that can lead to stroke and death. Recent studies have uncovered that AF-associated risk variants are adjacent to the Pitx2 locus, furthermore, the have demonstrated that AF patients with these variants respond better to certain antiarrhythmic drugs (AADs). However, the mechanisms underlying Pitx2 modulated arrhythmogenesis and variable effectiveness of AADs remain unclear. To untangle this issue, we have developed multi-scale computer models, ranging from single cell to tissue level, to mimic control and the Pitx2-knockout atria by incorporating experimental data on Pitx2-induced electrical and structural remodelling, as well as the effects of class I ADDs (flecainide). We discovered that removal of Pitx2 caused AF characterized by focal beats and re-entrant waves. It was found that these abnormalities can be attributed to triggered beats, action potential duration shortening and slow conduction arising from Pitx2-induced remodelling. Interestingly, these Pitx2-induced AF can be suppressed in the presence of class I AADs such as flecainide. Our simulations revealed that flecainide prevents triggered beats by blocking the function of ryanodine receptors and that flecainide can terminate re-entrant waves by prolonging action potential duration due to its actions on potassium channels.

## Introduction

Atrial fibrillation (AF), the most common sustained heart rhythm disorder, affects more than 33 million people worldwide and represents a growing cause of stroke and morbidity [1, 2]. Although the prevalence of AF increases with age and with the context of concomitant cardiac pathologies such as myocardial ischemia, hypertension and heart failure, genome-wide association studies (GWAS) have shown that one-third of AF patients carry common genetic variants, suggesting that AF has a heritable component[3–5]. These single nucleotide polymorphisms rs2200733 and rs10033464 were firstly identified in European, Chinese and Japanese populations on chromosome 4q25[6]. The gene-poor 4q25 region harbors the Pitx2 homeobox gene, which has been implicated in AF predisposition[3, 4, 7].

Pitx2 plays an important role in a left-sided signalling pathway that establishes the left-right asymmetry of the heart[8, 9]. Pitx2 encodes 3 isoforms (Pitx2a, Pitx2b and Pitx2c) and the Pitx2c isoform promotes the embryonic development of cardiac left–right asymmetry, with levels in right and left atriums being 1:100[10]. The atrial-selective transcription factor Pitx2 is an upstream transcriptional regulator of atrial electric function and cardiogenesis[11, 12]. Complete loss of Pitx2 function can result in malformation of the pulmonary veins that are well-known sites for ectopic activity promoting spontaneous AF[9]. Pitx2 loss-of-function mouse mutants displayed abnormal electrocardiograms with atrioventricular block, irregular R-R intervals and low voltage P waves[13, 14]. In the Pitx2-mutant atrial myocytes, a significantly more depolarized resting membrane potential (RMP)[14, 15], action potential duration (APD) shortening[10, 16, 17], and abnormalities in calcium handling[18, 19] were observed. Furthermore, expression array analyses identified genes related to calcium handling, gap junctions and ion channels affected by the reduced expression of Pitx2, mediating AF risk in carriers of common gene variants[10, 13, 14, 16, 19–21]. However, the precise AF pathophysiology under reduced Pitx2 remains unclear. Population-based studies have assessed the influence of common SNPs related to AF on the response to antiarrhythmic drug (AAD) therapies and showed that carriers of the variant allele at rs10033646 on chromosome 4q25 (Pitx2) responded favourably to the class I AAD (flecainide)[22–26]. The possible reasons behind this remain elusive.

In this study, we aimed to utilize virtual human atrial models to investigate the functional role of Pitx2-induced remodeling in atrial arrhythmogenesis and to examine the mechanism underlying the efficacy of flecainide for patients with Pitx2-induced AF. To achieve this, we adopted the previously modified and validated Courtemanche-Ramirez-Nattel model (CRN-TPA) for the human atrial cell by incorporating formulations of intracellular calcium dynamics from the ten Tusscher-Panfilov model to reproduce triggered activity[24, 27]. We then modified the model to produce four distinct models of Pitx2-induced electrical remodeling based on grouping existing literature. Furthermore, regional cellular heterogeneity between RA and LA, as well as Pitx2-induced structural remodeling including cell-cell coupling and fibrosis, was integrated into the computer models. Lastly, the antiarrhythmic action of flecainide was systematically simulated by integrating the block effects of flecainide on the sodium channel $I_{Na}$, rapidly activating delayed rectifier potassium channel $I_{Kr}$ and ryanodine receptor ($RyR$) into atrial cell models.

## Methods

An overview of our multi-scale human atrial models are provided in **Fig 1**. The downregulation of Pitx2 in membrane effector genes (the magenta circles) and alterations in ion channels and gap junction encoded by these genes were integrated into the multi-scale atrial models (**Fig 1A**). The developed models included single cells, 1D atrial strands and 2D atrial tissue. These Pitx2-mutant models were then incorporated with the actions of flecainide on ion channels (the red circles) to assess its efficacy.

### Human atrial cell model

At the single cell level, a recent human atrial cellular kinetics model (CRN-TPA) (**S1 Fig**) developed by our group was further adapted to simulate control and Pitx2 deficiency-induced AF conditions[24]. Cellular heterogeneity between LA and RA was modeled by taking into account differences in mRNA level for Pitx2 and in $I_{Kr}$ current density (**S1 Table**). Under control conditions, Pitx2 level in human LA is almost 100-fold higher in the LA as compared to that in RA[10] (**Fig 1B**) and $I_{Kr}$ current density in LA cells is 1.6-fold of that in RA cells[28, 29] (**Fig 1C**).

### Pitx2-induced electrophysiological remodeling

To study the effect of the Pitx2-dependent gene regulatory network in human atrial myocytes, we incorporated alterations in the ion channel properties due to Pitx2 deficiency into the CRN-TPA model. These Pitx2-induced changes in ion channels and cell-to-cell coupling have been characterized in many animal studies. Although these studies have shown SR calcium overload, RMP elevation and APD abbreviation in Pitx2-mutant mice atrial myocytes[10, 14–16, 18, 19], the identified remodeled ion channels are different in these studies: remodeling in $I_{K1}$ was identified among some studies[14, 15]; whereas remodeling in other channels, such as $I_{CaL}$, $I_{Ks}$ and $I_{Na}$, are present in others[16, 19–21]. Recent studies also characterized changes to the subcellular calcium handling properties and alterations to the tissue structure (i.e., fibrosis and gap junctions)[14, 21]. To incorporate such variations in experimental data into the computer models, four different scenarios (Pitx2-1, Pitx2-2, Pitx2-3 and Pitx2-4) (**Fig 1D**) were considered here for simulating Pitx2-induced remodeling (**Table 1**). Pitx2-1 included the remodeled inward-rectifier potassium current ($I_{K1}$) (green)[15], while Pitx2-2 included the remodeled L-type calcium current ($I_{CaL}$) and slow delayed rectifier potassium current ($I_{Ks}$) (magenta)[16]. Pitx2-3 has incorporated with remodelled key regulators associated with calcium handling, i.e., $I_{CaL}$, RyR and calcium transport ATPase (SERCA) (red)[19]. The Pitx2-4

**Multi-scale Human Atrial Model**

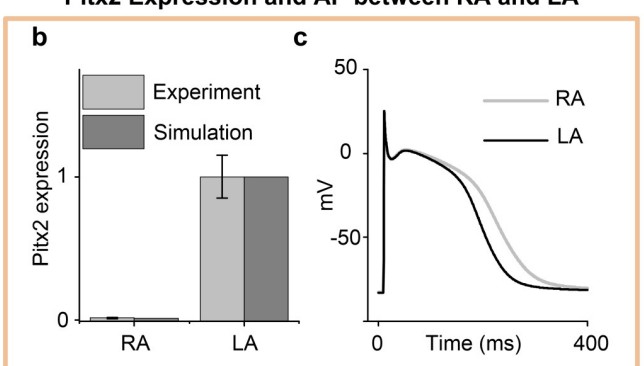

**Pitx2 Expression and AP between RA and LA**

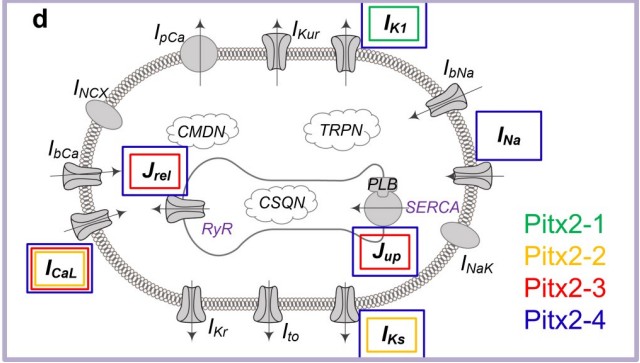

**Pitx2 Deficiency-Induced Human Atrial Cell Models**

**Fig 1. Multi-scale computer models to investigate the mechanism underlying Pitx2-induced AF and the effects of class Ic AAD (flecainide). a,** The computer models incorporated Pitx2-induced electrical and structural remodeling, and flecainide interactions with ion channels. Remodeled targets (magenta circles) included gap junction, $I_{K1}$, $I_{Ks}$, $I_{Na}$, $I_{CaL}$, RyR and SERCA. Drug targets of flecainide (red circles) were $I_{Kr}$, $I_{Na}$ and RyR. Under the control condition, heterogeneity in Pitx2 expressions (**b**) and AP (**c**) between LA and RA was included in the computer models. **d,** Based on experimental and clinical studies up to date, four Pitx2 deficiency-induced human atrial cell models were developed: Pitx2-1 with remodeled $I_{K1}$ (green), Pitx2-2 with remodelled $I_{Ks}$ and $I_{CaL}$ (magenta), Pitx2-3 with remodelled $I_{CaL}$, RyR and SERCA (red) and Pitx2-4 with remodelled $I_{K1}$, $I_{Ks}$, $I_{Na}$, $I_{CaL}$, RyR and SERCA (blue). Abbreviations: AF–atrial fibrillation; AAD–antiarrhythmic drug; LA–left atrium; RA–right atrium; AP–action potential; RyR–ryanodine receptor; SERCA–calcium transport ATPase.

**Table 1. Review of Pitx2-insufficiency induced remodelling data and model parameters in human left atrium.**

| Process | Experimental observation | Control | Pitx2-1 | Pitx2-2 | Pitx2-3 | Pitx2-4 |
|---|---|---|---|---|---|---|
| $I_{Na}$ | -60% (SCN5A & SCN1B) (Chinchilla et al., 2011); -40% (SCN5A & SCN1B) (Lozano-Velasco et al., 2017); +95% SCN5A (Nadadur et al., 2016); No Change (Syeda et al., 2016) | — | — | — | — | +10% |
| $I_{Ks}$ | +150% KCNQ1(Tao et al., http://circgenetics.ahajournals.org/2014); +100% (Pérez-Hernández et al., 2015); Increased volt-dependent potassium current (Kirchhof et al., 2011) | — | — | +100% | — | +150% |
| $I_{K1}$ | -20% (KCNJ2 & KCNJ12) (Chinchilla et al., 2011); +30 (KCNJ2 & KCNJ12) (Lozano-Velasco et al., 2017); -25% (Syeda et al., 2016) | — | -25% | — | — | -30% |
| $I_{CaL}$ | +500% CACNA1D (Tao et al., 2014); -50% (Pérez-Hernández et al., 2015); -50% (Lozano-Velasco et al., 2015); -30% CACNA1C (Lozano-Velasco et al., 2017); Decreased CACNA1C (Kirchhof et al., 2011) | — | — | -50% | -50% | -30% |
| SERCA | +50% ATP2A2(Tao et al., 2014); +1000% ATP2A2(Lozano-Velasco et al., 2015); +100% ATP2A2(Lozano-Velasco et al., 2017); +12% ATP2A2 (Nadadur et al., 2016) | — | — | — | +200% | +100% |
| RyR | +145% RyR2 (Tao et al., 2014); +30% RyR2 (Lozano-Velasco et al., 2015); +30% RyR2 (Lozano-Velasco et al., 2017); +10% RyR2 (Nadadur et al., 2016) | — | — | — | +30% | +30% |
| Gap junctions | -55% GJA1(Chinchilla et al., 2011); -5% GJA1(Nadadur et al., 2016); +100% GJA1 (Tao et al., 2014); +1000% (Pérez-Hernández et al., 2015); -50% (Pérez-Hernández et al., 2015); -58% (Kirchhof et al., 2011) | — | -50% | -50% | -50% | -50% |

Note: The remodelling in human right atrium is %1 of that in human left atrium. Abbreviations: RyR–ryanodine receptor; SERCA–calcium transport ATPase.

model took into account all identified regulators and best represented Pitx2 deficiency-induced electrical remodeling by including remodeled $I_{Na}$, $I_{Ks}$, $I_{K1}$, $I_{CaL}$, *RyR* and SERCA[15, 16, 19, 21] (blue). Furthermore, based on the assumption that the extent of electrical remodelling in atrial cells is dependent on mRNA level for Pitx2, Pitx2-modulated targets in LA cells have a 100 times greater change compared to those in RA cells. This enabled us to: (1) consider a broad range of experimental data on identified ion channel remodeling; (2) investigate the effects of varying degrees of remodeling to reproduce phenomenon observed in animal experiments; and (3) ascertain mechanisms underlying Pitx2-induced AF.

## Actions of flecainide on ion channels

To investigate the anti-arrhythmic effects of the class Ic drug flecainide, we integrated the actions of flecainide on ion channels and RyR into the Pitx2-mutant computer models. According to experimental/clinical data[26, 30], modifications to ion channels provoked by flecainide were modeled by using the standard sigmoid dose-response curve (**S2 Fig**) parametrized with $IC_{50}$ and Hill coefficient ($nH = 1$). The values of $IC_{50}$ for inhibition of $I_{Na}$, $I_{Kr}$ and RyR open probability were 84, 1.5[30] and 55 μM[26], respectively.

## 1D multicellular models

To study the effect of Pitx2-induced remodelling and flecainide on spatiotemporal behavior of electrical waves, we designed a 1D RA-LA strand model in which it has 75 RA myocytes and the other 75 LA cells. The diffusion coefficient (*D*) was set to a value of 0.1 cm²/s that gave a CV of a planar wave at 48.61 cm/s, within the physiological ranges (Slow: 30 to 40 cm/s, Normal: 60 to 75 cm/s, Fast: 150 to 200 cm/s). In Pitx2-mutant models, *D* in the LA was reduced by 50% to simulate the reduction in CV resulting from Pitx2-induced structural remodeling. Electrical waves in the strand model were evoked by the supra-threshold stimuli applied to three myocytes at the RA end. VW of unidirectional conduction block, an index to quantify the temporal vulnerability of cardiac tissue to re-entry, was quantified by varying the S1-S2 interval in the 1D homogeneous atrial strand. The protocol included 10 S1 stimuli applied at the end of the atrial strand and an S2 stimulus applied at the central segment of the atrial strand. Furthermore, VW$_{RA-LA}$ of unidirectional conduction block was measured to quantify

the RA-to-LA electrical heterogeneity[31, 32] by varying the S1-S2 interval in the RA-to-LA strand model with 250 RA myocytes and the other 250 LA cells.

## 2D multicellular models

To illuminate the spatiotemporal dynamics of triggered activity, a 500×500 square tissue model with randomly distributed normal myocytes, Pitx2-4 cells and fibrosis was designed to simulate synchronization of triggered beats and investigate the conditions of arrhythmia induction. Five different scenarios (#1, #2, #3, #4 and #5) were considered here for investigating effects of Pitx2-4 cells, fibrosis and cell-to-cell uncoupling on the inducibility of ectopic activity. D was set to be 100% for the #1, #2 and #3 conditions, and 30% for the #4 and #5 conditions, respectively. The ratios of normal cells, Pitx2-4 cells and fibrosis were set to be 80:20:0, 40:60:0, 78:20:2, 80:20:0 and 78:20:2, for the #1, #2, #3, #4 and #5 conditions, respectively. AP synchronization in the tissue model was simulated by a stimulus applied to the entire tissue at the beginning according to de Lang's method[33]. In addition, the spatiotemporal behavior of spiral waves in the drug-free Pitx2 setting versus in the presence of flecainide was investigated in the 500×500 homogeneous LA tissue. The spiral wave was initiated by a standard S1-S2 protocol. The S1 was applied to the left surface to evoke a planar excitation wave propagating towards the right part. The S2 was applied to a local area of the tissue within the VW to evoke unidirectional propagation that can lead to re-entry.

## Code availability

The ionic models for RA and LA atrial cells (Control, Pitx2-1, Pitx2-2, Pitx2-3 and Pitx2-4) and mathematical models of drug-channel interactions are freely available from the repository CellML (http://models.cellml.org/workspace/5c7). The electrophysiology codes developed by our team and can be obtained from GitHub (https://github.com/aspirerabbit).

## Results

### Pitx2-induced triggered activity in the single cell models

The ionic mechanisms of Pitx2 deficiency-induced AF were investigated by examining calcium transient ($Ca_i$) and action potentials (APs) using the computer models of control and four different scenarios of Pitx2-deficiency (Pitx2-1, Pitx2-2, Pitx2-3 and Pitx2-4) (**Fig 2**). At 1Hz pacing frequency, the Pitx2 deficiency-induced electrical remodeling produced augmented $Ca_i$ under Pitx2-3 and Pitx2-4 conditions, and delayed afterdepolarizations (DADs) under the Pitx2-3 condition (**Fig 2A**). Furthermore, we observed an increase in RMP (Pitx2-1 and Pitx2-4), large overshoot and maximum upstroke velocity ($dVdt_{max}$) (Pitx2-4), and APD abbreviation (Pitx2-2, Pitx2-3 and Pitx2-4) in Pitx2-deficient LA cells, compared with controls (**Fig 2B–2E**). However, these changes were absent in Pitx2-deficient RA cells (grey color). Thus, the augmented alterations in APs between RA and LA in the Pitx2-deficient settings, particularly in Pitx2-3 and Pitx2-4, significantly increased regional electrical heterogeneity. Additional simulations at 2Hz pacing frequency were performed to investigate the effects of pacing frequency on APs and $Ca_i$. The amplitude of $Ca_i$ was increased under all four conditions and triggered APs occurred in Pitx2-deficient LA cells (Pitx2-3 and Pitx2-4) when pacing frequency was increased (**Fig 2F–2J**).

To further evaluate the contribution of each remodelled target to the abnormalities in AP under Pitx2-4 condition, we conducted computer simulations by incorporating each ionic remodelling of Pitx2-4. Compared with the control AP, remodelled $I_{K1}$ resulted in a more positive RMP, increased $I_{Na}$ contributed to greater overshoot and $dVdt_{max}$, altered $I_{Ks}$, $I_{CaL}$ or

**AP and Ca$_i$ between RA and LA at a Pacing Frequency of 1Hz**

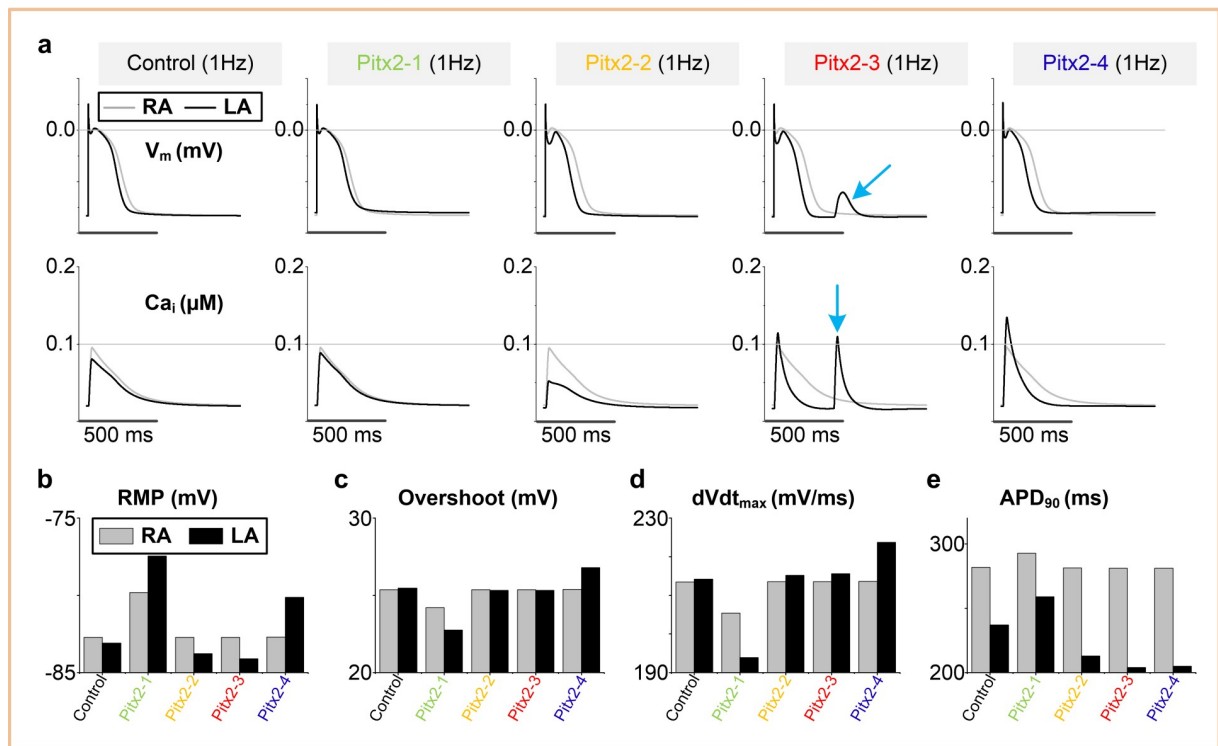

**AP and Ca$_i$ between RA and LA at a Pacing Frequency of 2Hz**

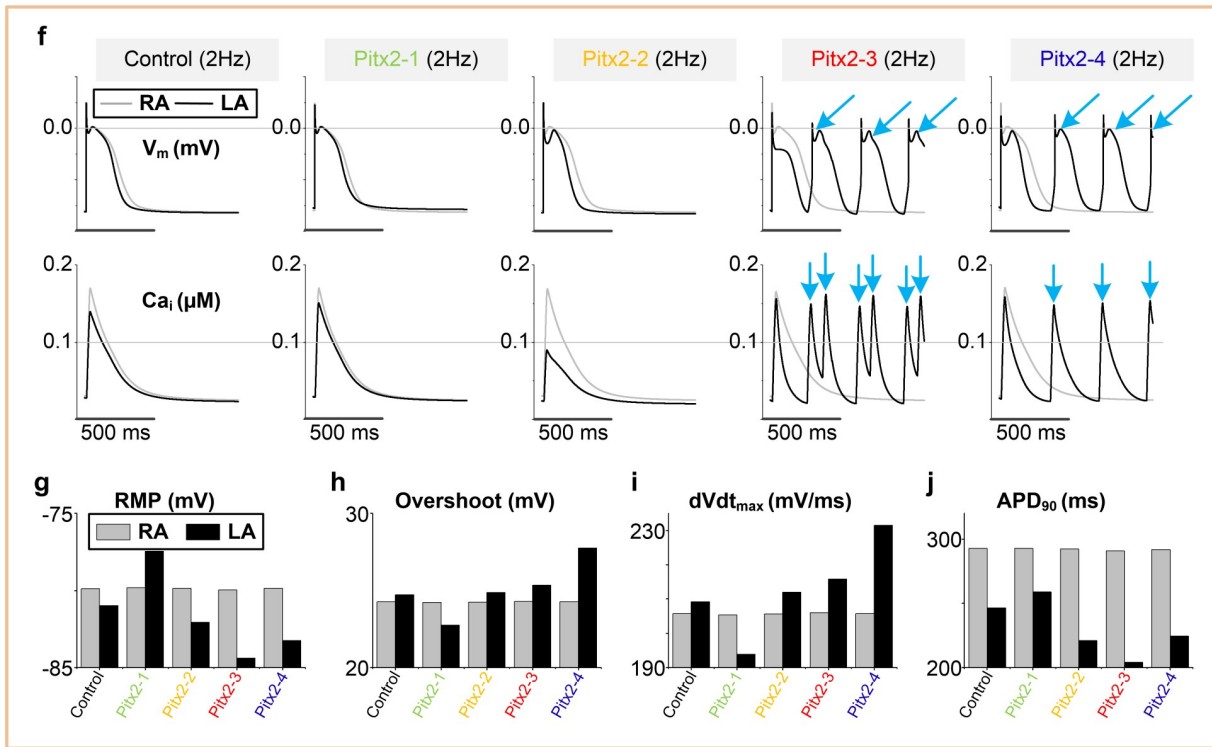

**Fig 2. Simulated action potentials (AP, V$_m$) and calcium transients (Ca$_i$) of left atrial (LA) and right atrial (RA) cells under controls and Pitx2-induced remodelling conditions.** At a pacing frequency of 1Hz, AP and Ca$_i$ (**a**), RMP (**b**), overshoot (**c**), dVdt$_{max}$ (**d**) and APD (**e**) under control, Pitx2-1, Pitx2-2, Pitx2-3 and Pitx2-4 conditions. Black and grey markers were used for LA and RA cells, respectively. Similar key

indicators (**f**-**j**) at a pacing frequency of 2Hz were displayed. Blue arrows indicated delayed afterdepolarizations, triggered action potentials and spontaneous calcium transients. Abbreviations: RMP–resting membrane potential; dVdt$_{max}$–maximum upstroke velocity; APD–action potential duration.

SERCA caused a reduction in APD$_{90}$, and enhanced SERCA increased sarcoplasmic reticulum calcium content (CaSR) and resulted in triggered APs (**S3A–S3F Fig**).

To further explore the putative targets among the remodelled cellular components that contribute to the Pitx2-4 phenotype, a series of simulations with modified Pitx2-4 models were performed by reversing each remodelled target separately. Compared with triggered APs in Pitx2-4 atrial cells, reversing SERCA remodelling rescued spontaneous depolarizations, but not in other conditions (**S3G Fig**).

### Effects of flecainide on Pitx2-induced arrhythmias at the single cell level

To assess the class Ic AAD flecainide, we have performed computer simulations with modelled the interaction of flecainide with $I_{Na}$, $I_{Kr}$ and RyR. **Fig 3** shows APs of Pitx2-deficient LA cells in the absence or presence of 2μM flecainide. In all cases of Pitx2-deficient LA myocytes, the flecainide therapy rescued abnormal depolarizations (Pitx2-3 and Pitx2-4), RMP elevation, increased overshoot, dVdt$_{max}$ acceleration, and APD abbreviation. To investigate exactly how the flecainide therapy contributes to the antiarrhythmic effects, we conducted additional simulations by including flecainide actions on each ionic target (i.e., $I_{Na}$, $I_{Kr}$ or RyR) separately in the Pitx2-4 model. Compared with APs in the presence of 2μM flecainide on all targets (**Fig 3F**), triggered APs were still observed in Pitx2-4 cells with the action of flecainide on $I_{Na}$ or $I_{Kr}$ only (**Fig 3G and 3H**) but not in LA myocytes with the effect of flecainide on RyR alone (**Fig 3I**).

Further simulations were conducted to examine whether the flecainide therapy can reduce the heterogeneity in AP features between RA and LA caused by Pitx2 deficiency-induced electrical remodelling. In presence of 2μM flecainide, our results indicate that heterogeneity in AP features (including AP profile, RMP, overshoot, dVdt$_{max}$ and APD$_{90}$) between RA and LA increased to varying degrees, compared to those in the absence of flecainide (**S4A–S4E Fig**). With the increase of flecainide concentration within the therapeutic range (0.5–2 μM), dVdt$_{max}$ decreased and APD$_{90}$ increased in both RA and LA cells. However, their differences between RA and LA cells were not reduced (**S4F and S4G Fig**).

### Arrhythmogenesis of Pitx2 deficiency in 1D computer model

We then examined the effect of Pitx2-induced remodeling on electrical excitation and heterogeneity of conduction in a 1D strand model with a total of 150 atrial cells (#1 - #75 for RA and #76 - #150 for LA). Under the control condition, pacing at the RA end (cell #1) using an extra stimulus led to electrical propagation from RA to LA. AP and Ca$_i$ along the control RA-LA strand had no obvious difference, while significant heterogeneities in AP and Ca$_i$ and triggered APs originating from LA were observed under Pitx2-deficient conditions (Pitx2-3 and Pitx2-4) (**Fig 4A**). After administration of 2 μM flecainide, these heterogeneities in the Pitx2-deficient strand still existed, but atrial ectopic beats were suppressed (**Fig 4B**). Similar to the single cell modelling results, these ectopic beats remained in Pitx2-4 cells with the action of flecainide on $I_{Na}$ or $I_{Kr}$ only, but were suppressed by the block effect of flecainide on RyR alone (**S5A–S5D Fig**).

To further quantify potential substrates of re-entrant arrhythmias, we measured conduction velocity (CV) and wavelength (WL) in both RA and LA. Compared with those under the control condition, reduction in CV and WL was observed in the four Pitx2-deficient LA strand

**Effect of 2 μM Flecainide on AP of Pitx2 Deficiency-Induced LA Cells**

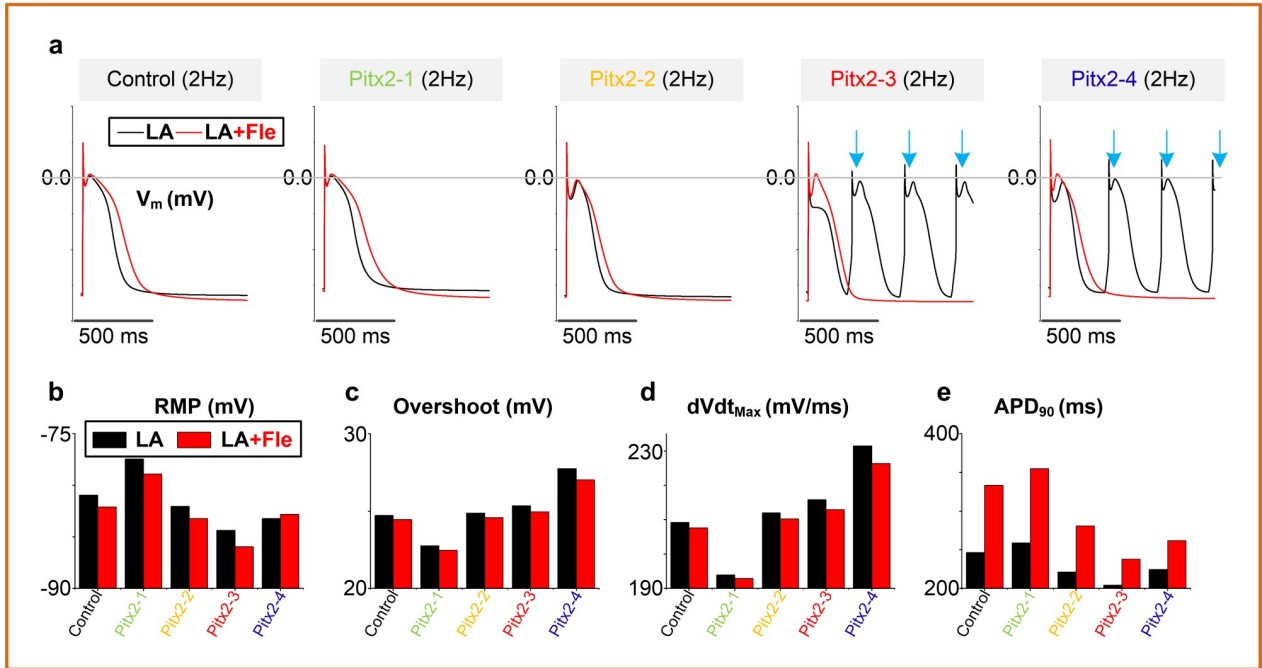

**Effects of Flecainide Action on AP in Pitx2-4 LA Cells**

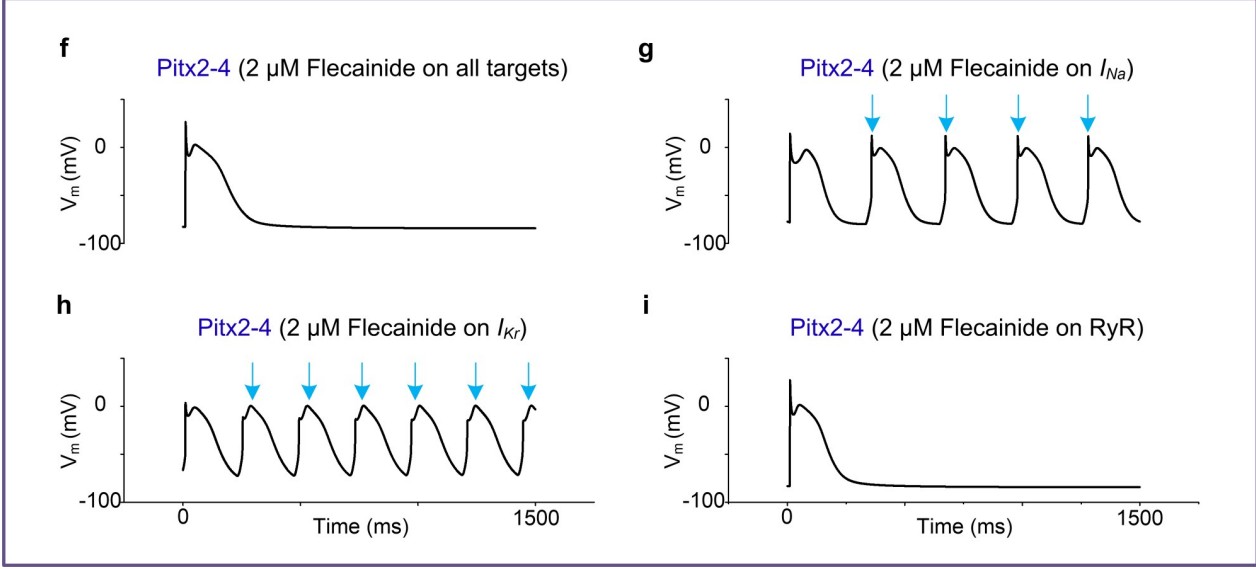

**Fig 3. Antiarrhythmic effects of flecainide on action potentials (AP, $V_m$). a,** Comparison of APs of LA cells in the presence or absence of 2 μM Fle. The main AP parameters included RMP (**b**), overshoot (**c**), dVdtmax (**d**) and APD (**e**). Simulated effects of 2 μM Fle on all targets (**f**), and on $I_{Na}$ (**g**), on $I_{kr}$ (**h**) and on RyR (**i**) respectively. Blue arrows indicate delayed afterdepolarizations. Abbreviations: LA–left atrium; Fle–flecainide; RMP–resting membrane potential; dVdt$_{max}$–maximum upstroke velocity; APD–action potential duration; RyR–ryanodine receptor.

models but not in the RA model (**Fig 4C and 4D**), leading to increased dispersion of repolarization and elevated susceptibility to re-entry. After administration of flecainide, for the control and four Pitx2-deficient LA cases, CV and the differences between RA and LA remained (**Fig 4E** and **S5E Fig**), while prolongation of WL occurred in both the LA and RA (**Fig 4F** and **S5F Fig**), thereby reducing the incidence of arrhythmias induced by Pitx2 deficiency.

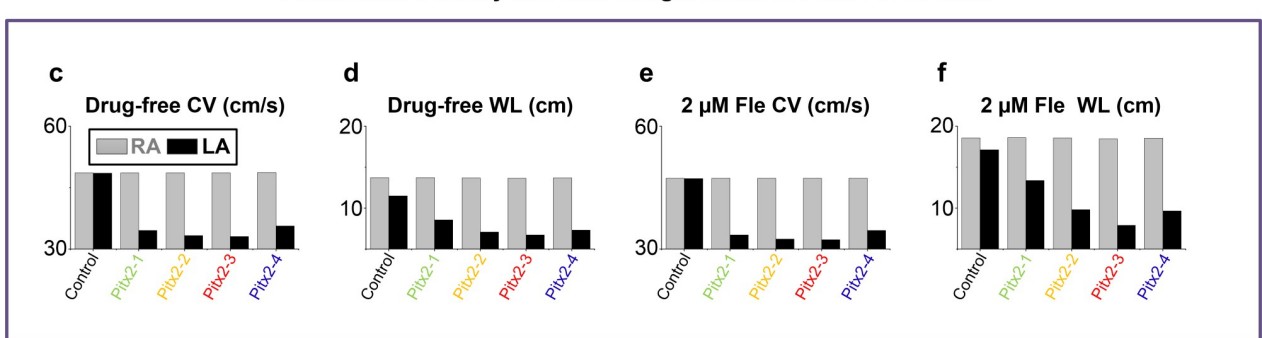

**Electrical and Calcium Waves in a RA-LA Strand**

**Conduction Velocity and Wavelength in the RA and LA Strands**

**Fig 4. Simulated electrical (V$_m$) and calcium (Ca$_i$) waves in a 1D RA-LA strand. a,** Simulated V$_m$ and Ca$_i$ waves in the drug-free settings (**a**) and in the presence of 2 μM flecainide (**b**). Comparison of CV(**c**) and WL (**d**) of LA (black) versus RA (grey) strands in the drug-free settings. In the presence of 2 μM flecainide, CV (**e**) and WL (**f**) are compared between RA and LA strands. The 1D strand contains 75 RA (#1-#75) cells and 75 LA (#76-#150) cells. Electrical waves were elicited by an extra stimulus at the RA end and propagated from RA to LA. Blue arrows indicate spontaneous delayed afterdepolarizations, triggered action potentials and calcium transients. Abbreviations: 1D –one-dimensional; RA–right atrium; LA–left atrium; CV–conduction velocity; WL–Wavelength.

## Structural remodelling promoting abnormal electrical activity at the tissue level

To further investigate focal arrhythmia arising from spatiotemporal synchronization of triggered APs due to Pitx2 deficiency as well as structural remodelling (fibrosis and cell-to-cell coupling), we designed a 2D tissue model (500×500) in which normal LA myocytes (white), Pitx2-4 remodeled LA cells (gray) and fibrosis (black) were randomly distributed (**Fig 5A**). The entire tissue was preconditioned at the beginning to produce AP synchronization according to de Lang's method[33]. In the well-coupled tissue, 20% Pitx2-4 cells could not overcome the sink-source mismatch to produce excitation waves (**Fig 5B**). However, increased number of Pitx2-4 cells, reduced cell-to-cell coupling or upregulated fibrosis (**S6 Fig**) can produce triggered waves (**Fig 5C–5E**). Furthermore, persistently triggered activity was observed in the model with both cell-to-cell uncoupling and fibrosis (**Fig 5F**). Thus, in addition to Pitx2 deficiency-induced electrical remodeling, structural remodeling can further increase susceptibility to ectopic beats at the tissue level.

We also investigated spiral wave dynamics and re-entry initiated from unidirectional conduction within the vulnerable window (VW) using an S1-S2 protocol. Under the drug-free Pitx2-4 condition, a spiral wave was initiated within its VW (230–232 ms) (**S7A Fig** and **S1 Video**) and transformed into fibrillated-like waves. Furthermore, it facilitated the development of triggered activity and interacted with ectopic beats (**Fig 5G**). In the presence of 2 μM flecainide, a planar wave slowly propagated and then produced a spiral wave within a shorter VW (294.7–296 ms) (**S7B Fig** and **S2 Video**). However, this re-entrant wave self-terminated at 8450 ms and no triggered activity occurred (**Fig 5H**). Thus, the efficacy of flecainide in suppressing Pitx2-induced AF may be attributed to its effects on triggered activity and WL.

A quantitative summary of electrophysiology characteristics is listed in **Table 2**.

## Discussion

Pitx2 plays a critical role in heart development and left-right atrial asymmetry, and its deficiency is associated with a 65% increased risk of AF[3]. Flecainide therapy for patients with Pitx2 deficiency seems promising, though its mechanism remains elusive[15, 25, 34]. To our knowledge, this is the first systematic in silico study to improve our understanding of the mechanism underlying Pitx2 deficiency induced AF by employing novel bio-physics based multi-scale computer models. More specifically, we have developed and studied a family of human atrial cellular models grouped by different levels of electrical remodelling due to Pitx2 deficiency, and at a single cell, 1D strand and 2D tissue level. Furthermore, we have included structural remodelling (fibrosis and cell-to-cell coupling) into our modelling study as well.

The key findings of this study are twofold. Firstly, we found that shortened APD, elevated RMP and slow conduction occur in LA cells with Pitx2 deficiency due to the identified remodelled vital channels. Notably, the elevated function of SERCA increases SR Ca$^{2+}$ concentration, thereby enhancing susceptibility to triggered activity. Furthermore, heterogeneity is further elevated due to Pitx2 deficiency: 1) Electrical heterogeneity between RA and LA increases; and 2) Increased fibrosis and decreased cell-cell coupling due to structural remodelling slow

## Structural Remodelling Increases Inducibility of Atrial Ectopic Beats

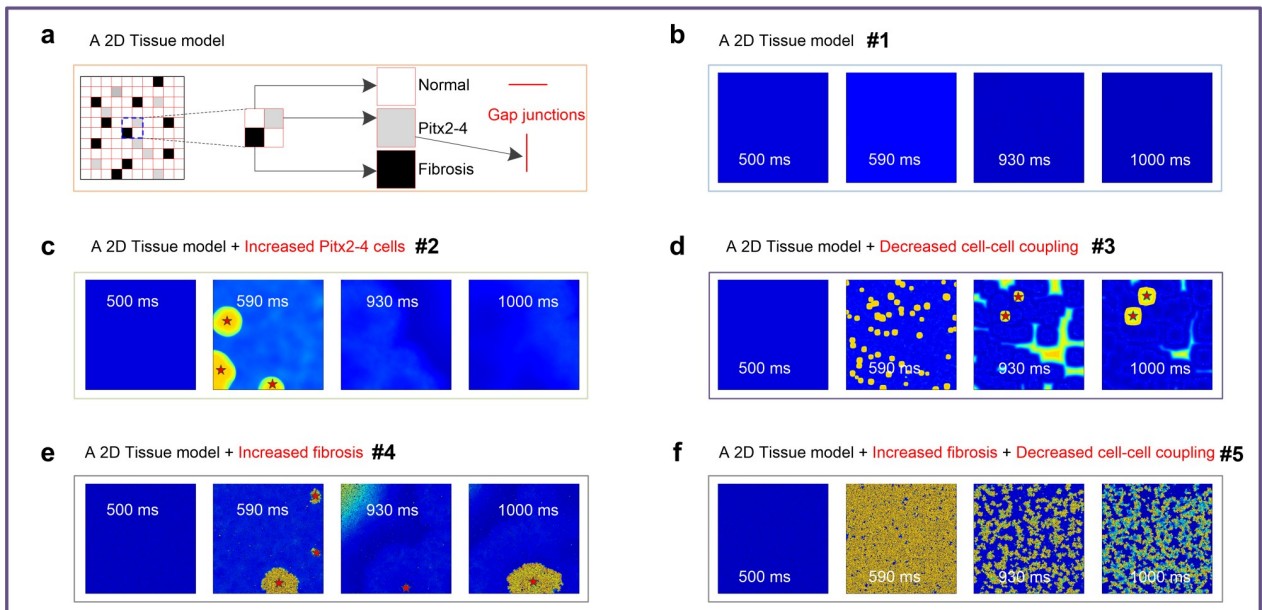

## Effects of Flecainide on Spiral Wave Dynamics in a Pitx2-4 Tissue Model

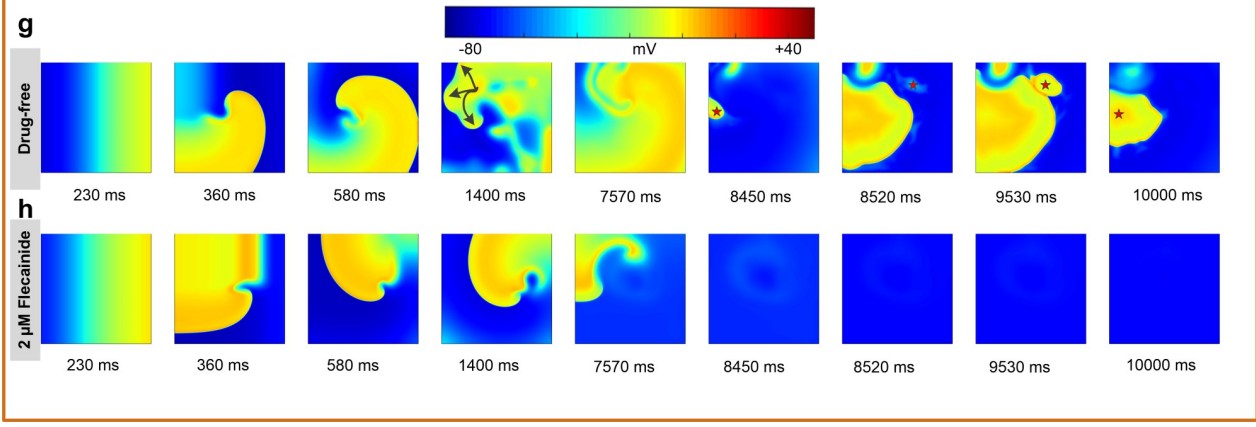

**Fig 5. Simulated spontaneous ectopic activity and re-entry. a,** A 500×500 square tissue model includes normal atrial myocytes, Pitx2-4 remodelled LA cells, fibrosis, and gap junctions. Simulated spontaneous ectopic activity in the tissue model with 20% Pitx2-4 cells (**b**), with further increased Pitx2-4 cells (**c**), with enhanced cell-to-cell uncoupling (**d**), with increased fibrosis (**e**), and with increased cell-to-cell uncoupling and fibrosis (**f**). Re-entrant waves in the drug-free Pitx2-4 settings (**g**) and in the presence of 2 μM flecainide (**h**). Note: For the #1 scenario, the diffusion coefficient (D) was set to be 100% and the ratio of normal cells, Pitx2-4 cells and fibrosis was set to be 80:20:0. For the #2 scenario, the number of Pitx2-4 cells was increased to 40% and thereby the ratio of different cell types was 40:60:0. And D was set to be 100%. For the #3 scenario, D was reduced to 30% to model cell-to-cell uncoupling and the ratio of different cell types was set to be 80:20:0. For the #4 scenario, fibrosis was increased to 2% and thereby the ratio of different cell types was 78:20:2. And D was set to be 100%. For the #5 scenario, fibrosis was increased to 2% and D was reduced to 30%. And the ratio of different cell types was 78:20:2.

electrical propagation and provide obstacles to attract re-entry, facilitating the initiation of re-entrant circuits. Secondly, our study suggests that the interaction of flecainide with RyR prevents spontaneous calcium release and reduces VW (or increases WL), contributing to the efficacy of the class Ic AADs. In contrast with existing studies, our study suggests that Na⁺ channel effect alone is insufficient to explain the effectiveness of flecainide. Additionally, the introduction of flecainide fails to reverse the RA-to-LA electrical heterogeneity.

**Table 2. A quantitative summary of electrophysiology characteristics.**

| | | | Control | Pitx2-1 | Pitx2-2 | Pitx2-3 | Pitx2-4 | Control+F | Pitx2-1+F | Pitx2-2+F | Pitx2-3+F | Pitx2-4+F |
|---|---|---|---|---|---|---|---|---|---|---|---|---|
| Cell | RMP | RA | -79.89 | -79.81 | -79.87 | -79.97 | -79.86 | -81.62 | -81.59 | -81.63 | -81.77 | -81.67 |
| | | LA | -80.97 | -77.45 | -82.05 | -84.38 | -83.23 | -82.10 | -78.90 | -83.23 | -85.94 | -82.82 |
| | OS | RA | 24.25 | 24.23 | 24.24 | 24.29 | 24.25 | 24.15 | 24.14 | 24.16 | 24.23 | 24.20 |
| | | LA | 24.73 | 22.75 | 24.86 | 25.33 | 27.73 | 24.45 | 22.45 | 24.56 | 24.94 | 27.01 |
| | dVdt$_{max}$ | RA | 205.7 | 205.3 | 205.6 | 206.0 | 205.7 | 205.6 | 205.5 | 205.6 | 206.2 | 205.9 |
| | | LA | 209.1 | 193.8 | 211.9 | 215.8 | 231.4 | 207.6 | 192.9 | 210.2 | 212.9 | 226.3 |
| | TA | RA | No | No | No | No | No | No | No | No | No | No |
| | | LA | No | No | No | Yes | Yes | No | No | No | No | No |
| | APD | RA | 292.5 | 292.7 | 292.0 | 290.6 | 291.5 | 358.9 | 359.1 | 358.3 | 356.8 | 357.5 |
| | | LA | 246.5 | 258.7 | 221.1 | 204.2 | 224.5 | 333.4 | 354.9 | 280.8 | 237.5 | 261.6 |
| | ΔAPD | | 46.0 | 34.0 | 70.9 | 86.4 | 67.0 | 25.5 | 4.2 | 77.5 | 119.3 | 95.9 |
| 1D | CV | RA | 48.61 | 48.62 | 48.60 | 48.59 | 48.63 | 47.24 | 47.27 | 47.25 | 47.23 | 47.28 |
| | | LA | 48.48 | 34.56 | 33.27 | 33.01 | 35.59 | 47.22 | 33.46 | 32.45 | 32.25 | 34.53 |
| | ΔCV | | 0.13 | 14.06 | 15.33 | 15.58 | 13.04 | 0.02 | 13.81 | 14.8 | 14.98 | 12.75 |
| | WL | RA | 13.69 | 13.70 | 13.66 | 13.65 | 13.66 | 18.57 | 18.59 | 18.54 | 18.44 | 18.50 |
| | | LA | 11.49 | 8.571 | 7.087 | 6.735 | 7.295 | 17.11 | 13.36 | 9.791 | 7.887 | 9.641 |
| | ΔWL | | 2.2 | 5.129 | 6.573 | 6.915 | 6.365 | 1.46 | 5.23 | 8.749 | 10.553 | 8.859 |
| | TA | | No | No | No | Yes | Yes | No | No | No | No | No |
| | VW$_{RA-LA}$ | | 293.8–294.5 | 324.8–327.8 | 293.8–295.8 | 291.8–298.8 | 291.8–294.8 | 359.8–360.8 | 398.8–402.8 | 332.8–349.8 | 332.8–352.8 | 332.8–349.8 |
| 2D | Re-entry | | - | - | - | - | Yes | - | - | - | - | No |
| | TA | | - | - | - | - | Yes | - | - | - | - | No |

Note: In order to evaluate the RA-to-LA electrical heterogeneity, VW$_{RA-LA}$ of unidirectional conduction block, an index to quantify the RA-to-LA electrical heterogeneity, was quantified by varying the S1-S2 interval in the RA-to-LA strand model with 250 RA myocytes and the other 250 LA cells. The protocol included 10 S1 stimuli applied at the end of the RA part and an S2 stimulus applied at the end segment of the atrial strand.

## Ionic basis of Pitx2 insufficiency-induced AP phenotype

Electrical remodelling due to Pitx2 insufficiency accounts for ectopic depolarizations[21, 24] and abbreviated APs[14], contributing to AF, but the ionic mechanisms underlying these changes in AP characteristics remain unclear. In the present study, we incorporated variations in experimental data into four computer models (Pitx2-1, Pitx2-2, Pitx2-3 and Pitx2-4). In these models, the Pitx2-4 model took into account all identified regulators and best represented Pitx2 deficiency-induced electrical remodelling. Ionic mechanisms underlying Pitx2 insufficiency-induced AP phenotype in different Pitx2 models were investigated by incorporating each ionic remodelling into the control model and reversing each remodelled target separately in the Pitx2-4 model. Consistent with experimental data on Pitx2 insufficiency-induced electrical remodelling[16, 19–21], remodelled targets considered in the Pitx2-4 model include $I_{K1}$, $I_{Na}$, $I_{CaL}$, $I_{Ks}$, RyR and SERCA. In these remodelled targets, we observed that downregulated $I_{K1}$ resulted in a more positive RMP and a prolonged APD, upregulated $I_{Na}$ contributed to a high overshoot and dVdt$_{max}$, the $I_{Ks}$ increase and (or) the $I_{CaL}$ decrease led to a reduction in APD, and increased SR calcium load due to enhanced SERCA function caused triggered APs. On the one hand, our simulated results support the notion that down-regulation of $I_{CaL}$ and up-regulation of $I_{Ks}$ are hallmarks of electrical remodelling in AF patients and mainly cause APD shortening[35, 36]. On the other hand, our results demonstrate that enhanced SERCA function can increase the incidence of spontaneous SR calcium release events, in line with the experimental observation in AF patients[37, 38]. The calcium extrusion due to the

spontaneous SR calcium release via $I_{NCX}$ can contribute to phase-4 depolarizations. In addition to the inward $I_{NCX}$ due to enhanced SERCA function, the remodelled $I_{K1}$ and $I_{Na}$ modulate RMP and membrane excitability[24], but our data and other studies[37, 38] suggest that triggered APs are mainly due to SR calcium load resulted from enhanced SERCA function.

## Role of remodelling under Pitx2-deficiency in arrhythmogenesis

Increased electrical heterogeneity between RA and LA due to down-regulated Pitx2 expression implicated in the initiation and maintenance of re-entrant arrhythmias.[39] Since the ratio of Pitx2 between LA and RA is 100:1 in the human atria,[10] we assumed that the extent of remodelling after Pitx2 deletion is dependent on the amount of Pitx2. In this study, increased electrical heterogeneity may result from the difference in Pitx2-induced remodelling between LA and RA. In our study, we observed that electrical remodeling induced by Pitx2 deficiency causes APD abbreviation and ectopic depolarizations in LA myocytes, but not in RA myocytes. Thus, Pitx2 insufficiency can cause an elevated difference in electrical properties between RA and LA cells, increasing repolarization dispersion in tissue and thereby susceptibility to the development of re-entry[32, 40, 41]. Furthermore, Pitx2 deficiency can also cause LA structural remodeling by regulating cardiac structural genes, increasing electrical and structural heterogeneity between the two atrial chambers. Finally, our computer simulation studies suggest that the introduction of flecainide can suppress triggered activity but fail to reverse the RA-to-LA electrical heterogeneity. The presence of increased fibrosis and decreased cell-to-cell coupling under structural remodelling further facilitates slow conduction, WL abbreviation, triggered activity and the initiation of re-entrant drivers, increasing susceptibility to AF. It is known that Pitx2-dependent network regulates cardiac structural genes (*Gja1*, *Gja5* and *Dsp*)[21] and Pitx2-induced structural remodeling leads to fibrosis and cell-cell uncoupling [42]. Also, structural remodeling due to the left-sided Pitx2 expression increases the intrinsic heterogeneity (i.e., CV and WL) between RA and LA, facilitating the development of re-entry (**Fig 6A**).

## Antiarrhythmic effects of flecainide

The class Ic AAD flecainide has antiarrhythmic effects on triggered activity by suppressing spontaneous SR calcium release via RyR, and on re-entrant arrhythmia by prolonging WL in Pitx2-induced AF. Population-based studies have established that the two SNPs, rs2200733 and rs10033464, from chromosome 4q25 near Pitx2 are associated with high incidence of AF [25]. Furthermore, there exists evidence that carriers of the variant allele at rs10033646 respond favourably to class Ic AADs including flecainide. The possible mechanism, as Syeda et al. suggest, is the inhibition of $I_{Na}$ in the presence of flecainide, therefore increasing post-repolarization refractory and suppressing arrhythmias in LA with Pitx2 deficiency.[15] Some experimental studies also suggest that the interaction of flecainide with $I_{Kr}$ and RyR plays an important role[26, 30]. The precise antiarrhythmic mechanism of flecainide on Pitx2-induced AF remains unsettled. In this study, our results provide novel mechanistic insights on the crucial role of actions of flecainide on $I_{Kr}$ and *RyR* in its antiarrhythmic effect. Our simulation study suggests inhibition of $I_{Na}$ alone in the presence of flecainide plays a minor role in suppressing AF. It is known that the action of flecainide on RyR can suppress triggered activity, for example, the previous studies on the flecainide therapy in catecholaminergic polymorphic ventricular tachycardia.[22, 26] Furthermore, flecainide had antiarrhythmic effects on Pitx2-induced AF due to its action on $I_{Kr}$ by prolonging WL, decreasing susceptibility of tissue to re-entrant arrhythmias (**Fig 6B**).

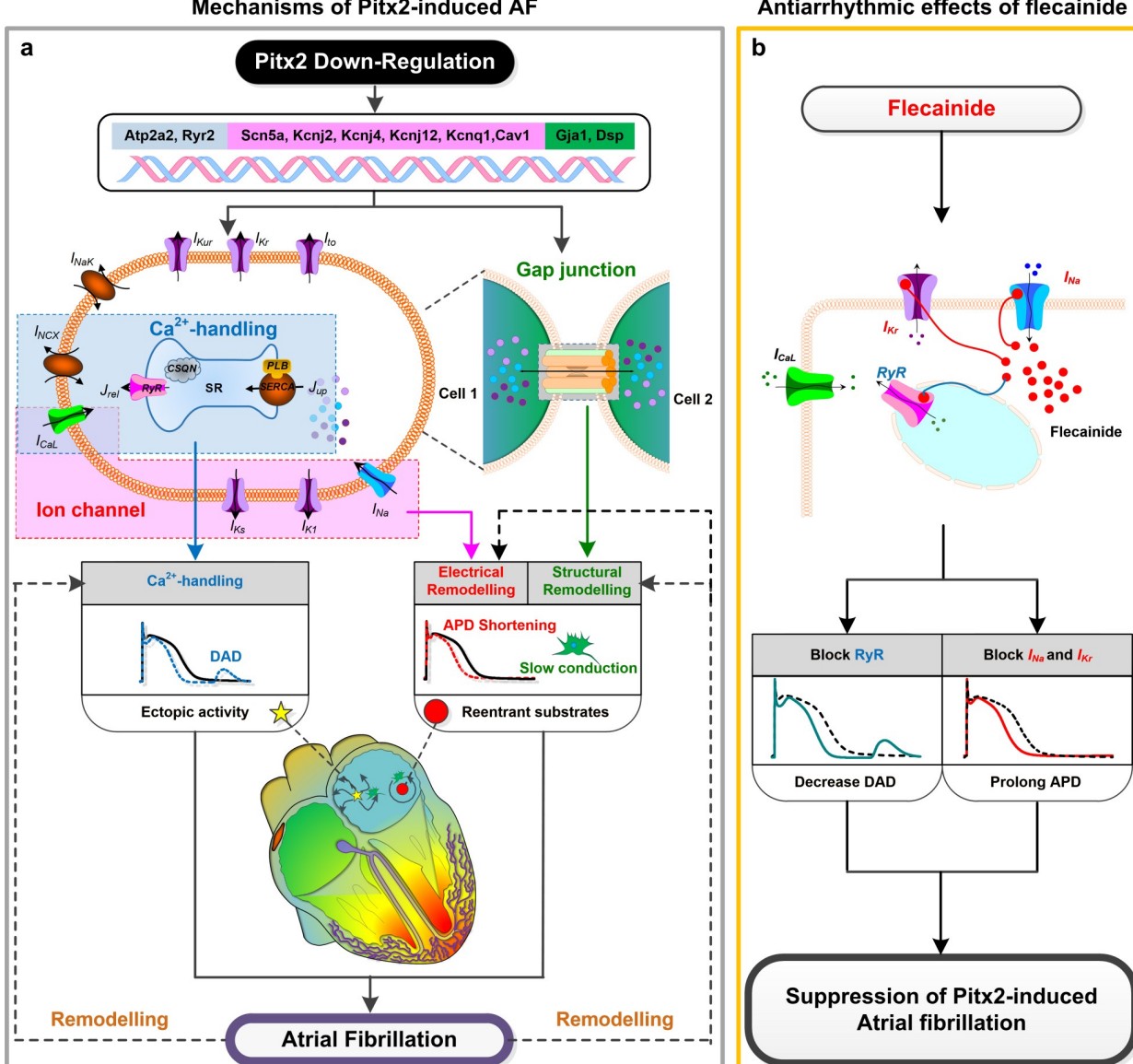

**Fig 6. Mechanisms underlying Pitx2 deficiency-induced AF and ionic mechanisms of anti-arrhythmic effects of flecainide in AF patients with Pitx2 down-regulation. a,** Pitx2 regulates calcium handling genes Atp2a2, Ryr2 and Sln, electrical remodelling genes Scn5a, Kcnj2, Kcnj4, Kcnj11, Kcnj12, Kcnq1 Cacna1d, Cacna2d2, and Cav1, and structural remodelling genes Gja1, Gja5 and Dsp. Pitx2 down-regulation in the LA leads calcium handling abnormities, electrical remodelling and structural remodelling, contributing to APD abbreviation, slowed atrial conduction and DAD, leading to AF triggers and substrates. **b**, Flecainide has block effects on RyR, $I_{Na}$ and $I_{Kr}$. Flecainide can reduce spontaneous calcium waves and triggered activity, and prolong the APD, thereby suppressing Pitx2 deficiency-induced AF. Abbreviations: AF–atrial fibrillation; LA–left atrium; DAD–delayed afterdepolarization; APD–action potential duration; RyR–ryanodine receptor.

However, flecainide also had adverse effects in Pitx2-induced AF. On the one hand, both of Pitx2-induced structural remodelling and the action of flecainide on $I_{Na}$ can slow CV and facilitate re-entry. On the other hand, both of Pitx2-induced electrical remodelling and the actions of flecainide on ion channels may increase the RA-to-LA electrical heterogeneity, promoting APD dispersion and thereby unidirectional conduction block indexed by VW$_{RA\text{-}LA}$[31, 32]. These findings support the notion that flecainide is not recommended in patients with structural heart disease due to high proarrhythmic risk based on the clinical findings[43]. Thus,

flecainide may be an effective antiarrhythmic drug for the treatment of Pitx2-induced AF patients without the structural disease[42, 44, 45].

## Limitations

Although the CRN model is based on human measurements, it cannot reproduce DADs due to overloaded calcium in the SR[46]. Therefore, the CRN model was modified and validated to generate our basal model (CRN-TPA) developed by our group[24]. In this study, the modified Grandi model and the CRN-TPA model were chosen as they are able to reproduce human AP morphology, triggered activity (**S8 Fig** and **S2 Table**), APD rate dependence and excitation dynamics for studying re-entrant arrhythmias in human atrial tissue[24, 47, 48]. There are several limitations special to this study here. Firstly, effects of Pitx2 insufficiency were assumed to be qualitatively similar between human and animal atria. Furthermore, in agreement with an experimental study[21] in which the extent of remodelling due to Pitx2 deletion was found to be dependent on the amount of Pitx2 expression, the ratio of remodelling between LA and RA was set to be 100:1. These assumptions warrant further investigations. Secondly, although we predicted Pitx2-induced AF phenotypes, including DADs in LA myocytes and ectopic beats and re-entry in LA tissues, these models do not explicitly represent subcellular calcium dynamics to simulate calcium sparks and the effects of the drugs on them[37, 38]. These ectopic beats in LA tissues were simulated by synchronizing LA cells and this protocol was based on experimental studies showing that spontaneous calcium releases are synchronous and overcome source-sink mismatch to generate focal arrhythmias in intact hearts[49–51]. Therefore, the mechanisms underlying spatiotemporal synchronization of SR calcium release in atrial tissue should be further explored. Thirdly, fibrosis was modelled as nonconducting tissue in this study as consistent with our previous studies[52, 53] and others[54]. In contrast, fibrosis was electrically coupled with healthy myocytes in some modelling studies[55]. Although different methodologies were used to model fibrosis in the past, they all draw the similar conclusions with regard to the role of fibrosis in AF[52, 56–58]. Finally, the precise spatial distribution of Pitx2 throughout human atria was unknown, therefore the regional difference in the cellular properties and remodelling was not investigated. Furthermore, we modelled atrial heterogeneity in the RA-LA tissue by considering the difference in $I_{Kr}$[28]. However, the ionic differences between the RA and LA are more diverse [59, 60], and this intrinsic heterogeneity should be considered. In addition, our idealized RA-LA tissue model did not include transition regions between RA and LA, electrical properties of transition regions, realistic geometry of these subregions, and fibre orientation. These limitations lead to a sharp transition between the RA and LA in our idealized tissue model. Special attention should be paid to explain our simulated results, and the effects of these factors on perpetuation and maintenance of re-entrant excitation should be further investigated[52]. Therefore, how the Pitx2-dependent gene regulatory network affects calcium homeostasis, intrinsic electrical heterogeneity, tissue structure, and atrial rhythm provide opportunities for further studies.

## Conclusions

Electrical and structural remodelling due to Pitx2 deficiency promoted arrhythmogenesis, leading to the development of after-depolarizations and re-entrant excitation. Effects of the class Ic AAD flecainide on AP morphology and tissue dynamics under Pitx2 deletion conditions, particularly its interaction with RyR for preventing spontaneous calcium release, demonstrated that flecainide is effective for the treatment of Pitx2-induced AF patients without structural diseases. We expect that these and analogous efforts will contribute to improved platforms for AF risk determination and therapeutic stratification.

## Supporting information

**S1 Fig. Intracellular structures of the Courtemanche et al. model (CRN), our human atrial (TPA) model and a new human atrial model (CRN_TPA) constructed by integrating the calcium dynamics of our TPA model into the CRN model.** The CRN_TPA model was developed by combining the calcium handling formulations from the TPA model and the transmembrane currents of the CRN model. The cell space includes a sub-cellular compartment dyadic cleft (SS), the sarcoplasmic reticulum (SR), the cytoplasm and cell membrane.
(DOCX)

**S2 Fig. Interactions of Class Ic AAD flecainide with ion channels.** Dose-response for the inhibitory effects of flecainide on $I_{Na}$, $I_{Kr}$, and RyR showed that IC50 values are 84±4 μM, 1.5 ±0.1 μM, and 55±8 μM, respectively. Abbreviations: AAD–antiarrhythmic drug; RyR–ryanodine receptor; $IC_{50}$ half-maximal inhibitory concentration.
(DOCX)

**S3 Fig. Role of individual remodelled targets in the Pitx2 deficiency-induced atrial fibrillation.** Effects of individual remodelled targets ($I_{K1}$, $I_{Na}$, $I_{Ks}$, $I_{CaL}$, RyR or SERCA) on AP (a), RMP (b), overshoot (c), dVdtmax (d), APD (e) and CaSR (f). Effects of reversing remodelled individual targets on key indicators (g-l). haBlue arrows indicate spontaneous delayed afterdepolarization (DAD). Abbreviations: RyR–ryanodine receptor; SERCA–calcium transport ATPase; AP–Action potential; RMP–resting membrane potential; dVdtmax–maximum upstroke velocity; APD–action potential duration; CaSR–sarcoplasmic reticulum calcium content.
(DOCX)

**S4 Fig. Antiarrhythmic effects of flecainide on action potentials (AP, Vm) of LA and RA cells under control, Pitx2-1, Pitx2-2, Pitx2-3 and Pitx2-4 conditions.** a, Comparison of APs of LA (red) versus RA (gray) cells in the presence of 2μM flecainide under control and four Pitx2-deficiency conditions. The main AP parameters included RMP (b), overshoot (c), dVdtmax (d) and APD (e). Within clinical dose (0.5~2 μM), flecainide reduced dVdtmax (f) and prolonged APD (g). Abbreviations: LA–left atrium; RA–right atrium; Fle–flecainide; RMP–resting membrane potential; dVdtmax–maximum upstroke velocity; APD–action potential duration.
(DOCX)

**S5 Fig. Antiarrhythmic effects of flecainide on electrical (Vm) and calcium (Cai) waves.** Compared with Vm and Cai waves in the drug-free Pitx2-4 settings (a), these waves in the presence of 2 μM flecainide on targeting $I_{Na}$ (b), on $I_{kr}$ (c) and on RyR alone (d) respectively. Blue arrows indicate spontaneous delayed afterdepolarizations, triggered action potentials and calcium transients. Within clinical dose (0.5~2 μM), flecainide reduced CV (e) and prolonged WL (f). Abbreviations: RyR–ryanodine receptor; CV–conduction velocity; WL–Wavelength.
(DOCX)

**S6 Fig. Effects of fibrosis and cell-to-cell uncoupling on ectopic beats.** a, Simulated spontaneous ectopic activity and re-entrant waves in the tissue model with increased fibrosis. b, Simulated ectopic activity and re-entrant waves in the tissue model with cell-to-cell uncoupling.
(DOCX)

**S7 Fig. Vulnerable window (VW) of unidirectional block from premature stimulation.** Bidirectional conduction block, unidirectional conduction block and bidirectional conduction in the drug-free Pitx2-4 settings (a) versus in the presence of 2 μM flecainide (b). VW under

Pitx2-1, Pitx2-2, Pitx2-3 and Pitx2-4 conditions in the drug-free Pitx2-4 settings (c) versus in the presence of 2μM flecainide (d).
(DOCX)

**S8 Fig. Simulated action potentials (AP) of left atrial (LA) and right atrial (RA) cells under controls and Pitx2-induced remodelling conditions.** At a pacing frequency of 2Hz, AP under control, Pitx2-1, Pitx2-2, Pitx2-3 and Pitx2-4 conditions. Black and grey markers were used for LA and RA cells, respectively. Blue arrows indicate spontaneous delayed afterdepolarizations and triggered action potentials.
(DOCX)

**S9 Fig. Effects of Pitx2-induced remodelling on action potential duration (APD) restitution properties.** (a-e) APD restitution curves for control, Pitx2-1, Pitx2-2, Pitx2-3 and Pitx2-4 conditions. Black and grey markers were used for LA and RA cells, respectively. Abbreviations: APD–action potential duration; DI–diastolic interval; LA–left atrial cell; RA–right atrial cell.
(DOCX)

**S10 Fig. Simulated action potentials (AP) of left atrial (LA) and pulmonary vein (PV) cells under controls and Pitx2-induced remodelling conditions.** At a pacing frequency of 2Hz, AP under control, Pitx2-1, Pitx2-2, Pitx2-3 and Pitx2-4 conditions. Black and red markers were used for LA and PV cells, respectively. Blue arrows indicate spontaneous delayed afterdepolarizations and triggered action potentials.
(DOCX)

**S1 Table. Ionic differences in regional cell models.**
(DOCX)

**S2 Table. A quantitative summary of electrophysiology characteristics for left atrial (LA) and pulmonary vein (PV) cells under controls and Pitx2-induced remodelling conditions.**
(DOCX)

**S1 Video. Re-entry in 2D idealized geometry under the drug-free Pitx2-4 condition.**
(AVI)

**S2 Video. Re-entry in 2D idealized geometry under the Pitx2-4 condition in the presence of 2 μM flecainide.**
(AVI)

**S1 Text. Supplementary Methods and Results.**
(DOCX)

## Author Contributions

**Conceptualization:** Jieyun Bai, Patrick A. Gladding, Jichao Zhao.

**Data curation:** Jieyun Bai.

**Formal analysis:** Jieyun Bai.

**Funding acquisition:** Martin K. Stiles, Vadim V. Fedorov, Jichao Zhao.

**Investigation:** Jieyun Bai, Patrick A. Gladding, Martin K. Stiles, Vadim V. Fedorov, Jichao Zhao.

**Methodology:** Jieyun Bai, Andy Lo.

**Project administration:** Jichao Zhao.

**Resources:** Jichao Zhao.

**Software:** Jichao Zhao.

**Supervision:** Jichao Zhao.

**Validation:** Jieyun Bai, Andy Lo, Jichao Zhao.

**Visualization:** Jieyun Bai.

**Writing – original draft:** Jieyun Bai.

**Writing – review & editing:** Andy Lo, Patrick A. Gladding, Martin K. Stiles, Vadim V. Fedorov, Jichao Zhao.

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
