## [Decision Letter · Decision Letter 0]

13 Sep 2019

Dear Dr Zhao,

Thank you very much for submitting your manuscript, 'Resolving the mechanism underlying increased susceptibility to atrial fibrillation in patients with impaired Pitx2', to PLOS Computational Biology. As with all papers submitted to the journal, yours was fully evaluated by the PLOS Computational Biology editorial team, and in this case, by independent peer reviewers. The reviewers appreciated the attention to an important topic but identified some aspects of the manuscript that should be improved.

We would therefore like to ask you to modify the manuscript according to the review recommendations before we can consider your manuscript for acceptance. Your revisions should address the specific points made by each reviewer and we encourage you to respond to particular issues Please note while forming your response, if your article is accepted, you may have the opportunity to make the peer review history publicly available. The record will include editor decision letters (with reviews) and your responses to reviewer comments. If eligible, we will contact you to opt in or out.raised.

- Supporting Information uploaded as separate files, titled 'Dataset', 'Figure', 'Table', 'Text', 'Protocol', 'Audio', or 'Video'.

We hope to receive your revised manuscript within the next 30 days. If you anticipate any delay in its return, we ask that you let us know the expected resubmission date by email at ploscompbiol@plos.org.

Sincerely,

Andrew D. McCulloch, Ph.D.

Associate Editor

PLOS Computational Biology

Daniel Beard

Deputy Editor

PLOS Computational Biology

[LINK]

Reviewer's Responses to Questions

**Comments to the Authors:**

Reviewer #1: This is an interesting modelling study of the role of genetic factors in the role of atrial fibrillation (AF). Specifically, the study focuses on mechanisms underlying Pitx2 deficiency-induced AF, as well as effects of anti-arrhythmic drug (AAD) flecainide in AF patients with Pitx2 down-regulation. Pitx2 deficiency promoted AF arrhythmogenesis in single-cell and tissue atrial models through to the development of after-depolarizations and re-entrant excitations, whereas flecainide prevents both. Overall, the work is novel and interesting, but the paper could benefit from clarification and a better presentation of some of the results. Please see my specific comments and suggestions below.

1) The paper claims that “the introduction of flecainide fails to reverse the RA-to-LA electrical heterogeneity”. However, it also stated that “in the presence of 2 μM flecainide, a planar wave … produced a spiral wave within a shorter VW” (compared to the flecainide-free case, by less than 1 ms). It has been shown in several studied that shorter VW is a sign of reduced electrical heterogeneity (Colman et al., 2014; 16: 416-23, Varela et al., 2016; 12: e1005245). Hence, I would expect that APD dispersion under flecainide should be reduced by 1-2 ms – was this measured? From the figures presented this is impossible to conclude, would it be possible to provide the actual numbers? It is also worth referencing the aforementioned studies, as they demonstrate the important role of increased atrial heterogeneity in AF arrhythmogenesis, and the modulating effects of AADs on the APD and APD dispersion. If flecainide in this study indeed reduces atrial heterogeneity even by a few ms, it should contribute to its anti-arrhythmic effect, as shown in the studies by Colman and Varela.

2) Four genetic variants Pitx2-1, -2, -3, -4 are considered in single-cell, 1D and 2D tissue models and multiple electrophysiology characteristics are measured in each case. Due do the large number of cases considered, it’s not easy to see the general picture. The paper could benefit from one or more tables summarizing these multiple characteristics and their inter-relations (e.g., APD and VW increased/decreased, triggered activity or re-entry present/absent, etc).

3) Atrial heterogeneity itself is modelled quite simplistically, which is a major limitation. First, there is a sharp transition between the RA and LA, which is non-physiological, since there is no direct conduction between the chambers (and even if there was, it would not have been a sharp charge). Moreover, RA-LA differences are accounted for by changing only one ionic current, IKr. However, ionic difference between the RA and LA are more diverse (Colman et al., J Physiol. 2013; 591: 4249-72). This should at least be mentioned in Limitations.

4) Whereas all four Pitx2 models are characterised by shorter APD in the LA compared to RA, other AP characteristics change differently between the chambers and Pitx2 models. For example, RMP is higher in the LA in Pitx2-1 and Pitx2-4, but higher in the RA in Pitx2-2 and Pitx2-3; moreover, these difference are rate-dependent (Figure 2). It’s worth discussing what are ionic mechanisms of such differences, and which Pitx2 model may be most physiological.

5) I think some phrasing throughout the paper can be improved. First, the title “Resolving the mechanism…” makes an excessively strong statement and should be toned downed. Similarly, a statement about “demystifying” the role of Pitx2 in Introduction should be moderated. Then the second para of Introduction uses a lot of terms and jargon specific to a narrow research field and is difficult to follow – please re-write using more general terms. Finally, in what sense are rs2200733 and rs10033464 “the most important variants”?

6) Figure 5d and f are difficult to interpret (f in particular). Are we looking at the voltage? Do all cells in panels f fire at once (going from blue to yellow) and then become desynchronised (mixture of blue and yellow in the last two panels)? These patterns look non-physiological. Why in panels d the patterns are different, with desynchrony seen already in the second panel? Please also see comment 2 above about better systemising all results in the paper.

Reviewer #2: This study by Bai et al. uses computational simulations to investigate the effects of impaired Pitx2 properties on atrial fibrillation susceptibility and reentry properties. The study uses single cell, 1D and 2D simulations to investigate these effects, comparing four different Pitx2 cell model representations. The effects of flecainide are then simulated on one of the cell model variants. This represents a nice and well-designed use of simulation to investigate AF mechanisms, incorporating available experimental data. I have a few main comments, and some minor comments for improving the presentation of the study.

Main comments:

1) The results of this study may be highly dependent on the choice of atrial cell model. The authors show that calcium dynamics are important in predicting the effects of pitx2. Please could the authors justify their choice of cell model, and if possible predict how their findings would change with the use of a different atrial cell model?

2) How do you think pulmonary vein cell response would compare to the left atrial cell model findings here? Could you perform a subset of simulations for a cell model with ionic properties modified to match pulmonary vein cells and see how DAD incidence and action potential properties change?

3) The parameter changes for the different Pitx2 cell models should be included in the main body of the text rather than just in the supplement as this is fundamental to the study.

4) It would be helpful to have more quantitative results in the text and not simply in the figures.

Minor comments:

Abstract:

• “Secondly, our study suggests...” It isn’t clear whether this finding is with or without Pitx2 changes, and which of the parameter sets it refers to.

Introduction:

• “The lack of Pitx2 can alter right-left atrium…” this sentence is very unclear.

Results:

• Are there are reported differences between Pitx2 level in the appendages or pulmonary veins compared to the atrial body?

• Section 3.1 is methods not results. It might be clearer to include the methods section first and then the results section.

• Why did you decide to use 1Hz and 2Hz pacing? Is 2Hz fast enough to test restitution properties? Please could you test the restitution properties of each cell model?

• The text says heterogeneity in AP features between RA and LA increased to varying degrees, and then later it says differences between RA and LA cells remained unchanged. Please clarify.

• Please include values in the text to support your statements. How much did APD decrease by etc.

• How did the Pitx2-3 cells respond to the block effect of flecainide?

• It would be very interesting to see how these differences in LA vs RA response affect AF simulations in a realistic bi atrial model.

• “Persistently triggered activity was observed in the model with both cell-to-cell uncoupling and fibrosis”. Could you quantify this in terms of frequency of triggers, amount of uncoupling and degree of fibrosis?

• Please comment on the magnitude of the change in vulnerable window size. It seems like a small change with flecanide?

Discussion:

• Key finding 2: Increased fibrosis and decreased cell-to-cell coupling…” Did you show these anchor reentry in this paper?

• High “accidence” of AF – should be “incidence”

• Limitations: Please discuss the choice of cell model.

• Limitations: The effect of fibrosis modeling methodology has been shown to affect simulated AF dynamics in previous studies.

Methods:

• Please include more details on how this model differs from the original CRN model and also on % changes in conductances for each Pitx2 model.

• Please include a reference for IC50 values.

• How much did CV change by for the reduced diffusion coefficient simulations?

• What % of cells were normal, Pitx2-4 and fibrosis in the simulations? Did you investigate the sensitivity of simulation outcome to this?

• Please include a link to your github.

Figure 1: Make subfigure d larger

Figure 4: Are any changes significantly different?

Figure 5: Not clear what b-f show?

**Have all data underlying the figures and results presented in the manuscript been provided?**

Reviewer #1: Yes

Reviewer #2: Yes

PLOS authors have the option to publish the peer review history of their article (what does this mean?). If published, this will include your full peer review and any attached files.

Reviewer #1: Yes: Oleg Aslanidi

Reviewer #2: No

---

## [Decision Letter · Decision Letter 1]

22 Jan 2020

Dear Dr Zhao,

We are pleased to inform you that your manuscript 'In silico investigation of the mechanisms underlying atrial fibrillation due to impaired Pitx2' has been provisionally accepted for publication in PLOS Computational Biology.

In the meantime, please log into Editorial Manager at https://www.editorialmanager.com/pcompbiol/, click the "Update My Information" link at the top of the page, and update your user information to ensure an efficient production and billing process.

One of the goals of PLOS is to make science accessible to educators and the public. PLOS staff issue occasional press releases and make early versions of PLOS Computational Biology articles available to science writers and journalists. PLOS staff also collaborate with Communication and Public Information Offices and would be happy to work with the relevant people at your institution or funding agency. If your institution or funding agency is interested in promoting your findings, please ask them to coordinate their releases with PLOS (contact ploscompbiol@plos.org).

Thank you again for supporting Open Access publishing. We look forward to publishing your paper in PLOS Computational Biology.

Sincerely,

Andrew D. McCulloch, Ph.D.

Associate Editor

PLOS Computational Biology

Daniel Beard

Deputy Editor

PLOS Computational Biology

Reviewer's Responses to Questions

**Comments to the Authors:**

Reviewer #1: The authors did a great job revising the manuscript and carefully addressing all the reviewers' comments. All points raised previously have been clarified and the new figures and tables are highly informative. I can only congratulate the authors.

Reviewer #2: Thank you for responding to all of my comments. Congratulations on an interesting and thorough study.

**Have all data underlying the figures and results presented in the manuscript been provided?**

Reviewer #1: Yes

Reviewer #2: Yes

PLOS authors have the option to publish the peer review history of their article (what does this mean?). If published, this will include your full peer review and any attached files.

Reviewer #1: Yes: Oleg Aslanidi

Reviewer #2: No

---

## [Editor Report · Acceptance letter]

20 Feb 2020

PCOMPBIOL-D-19-01121R1 

In silico investigation of the mechanisms underlying atrial fibrillation due to impaired Pitx2

Dear Dr Zhao,

I am pleased to inform you that your manuscript has been formally accepted for publication in PLOS Computational Biology. Your manuscript is now with our production department and you will be notified of the publication date in due course.

With kind regards,

Bailey Hanna
